# Magnetic Torus Microreactor as a Novel Device for Sample Treatment via Solid-Phase Microextraction Coupled to Graphite Furnace Atomic Absorption Spectroscopy: A Route for Arsenic Pre-Concentration

**DOI:** 10.3390/molecules27196198

**Published:** 2022-09-21

**Authors:** Sofía Ortegón, Paula Andrea Peñaranda, Cristian F. Rodríguez, Mabel Juliana Noguera, Sergio Leonardo Florez, Juan C. Cruz, Ricardo E. Rivas, Johann F. Osma

**Affiliations:** 1Department of Chemistry, Universidad de Los Andes, Cra. 1E No. 19a-40, Bogotá 111711, Colombia; 2Department of Electrical and Electronic Engineering, Universidad de Los Andes, Cra. 1E No. 19a-40, Bogotá 111711, Colombia; 3Department of Biomedical Engineering, Universidad de Los Andes, Cra. 1E No. 19a-40, Bogotá 111711, Colombia

**Keywords:** torus microreactor, magnetic solid microextraction, arsenic preconcentration, graphite furnace detection

## Abstract

This work studied the feasibility of using a novel microreactor based on torus geometry to carry out a sample pretreatment before its analysis by graphite furnace atomic absorption. The miniaturized retention of total arsenic was performed on the surface of a magnetic sorbent material consisting of 6 mg of magnetite (Fe_3_O_4_) confined in a very small space inside (20.1 µL) a polyacrylate device filling an internal lumen (inside space). Using this geometric design, a simulation theoretical study demonstrated a notable improvement in the analyte adsorption process on the solid extractant surface. Compared to single-layer geometries, the torus microreactor geometry brought on flow turbulence within the liquid along the curvatures inside the device channels, improving the efficiency of analyte–extractant contact and therefore leading to a high preconcentration factor. According to this design, the magnetic solid phase was held internally as a surface bed with the use of an 8 mm-diameter cylindric neodymium magnet, allowing the pass of a fixed volume of an arsenic aqueous standard solution. A preconcentration factor of up to 60 was found to reduce the typical “characteristic mass” (as sensitivity parameter) determined by direct measurement from 53.66 pg to 0.88 pg, showing an essential improvement in the arsenic signal sensitivity by absorption atomic spectrometry. This methodology emulates a miniaturized micro-solid-phase extraction system for flow-through water pretreatment samples in chemical analysis before coupling to techniques that employ reduced sample volumes, such as graphite furnace atomic absorption spectroscopy.

## 1. Introduction

The development of microfluidics, an emerging field of science and technology, focuses on the design and development of small devices that explore, from a miniaturized perspective and level, new scientific avenues for manipulating fluids and objects to obtain assemblies with different functionalities, monitor environmental or bodily analytes or perform chemical, pharmaceutical, and biological analyses [1,2]. As a result, this technology has found application in a variety of fields, including biomedicine [3,4], analytical biochemistry, microbiology [5], medical diagnostics, nanobiotechnology, and environmental monitoring and treatment [6,7]. In this regard, many of these devices have been used for detecting and quantifying environmental pollutants and heavy metals very precisely [8,9]. Conversely, numerous reports can be found in the literature related to miniaturized analytical devices for metals such as nickel, copper, cadmium, lead, mercury, and arsenic [10,11,12,13,14,15].

Microfluidic devices for analytical analyses offer several advantages, including simple pre-treatment sample procedures, significantly reduced amounts of samples and reagents, highly controlled mixing and separation processes, and in situ analyte detection due to the ease of use of the instrumentation and automation [16,17]. Compared with conventional analytical systems, this might also translate into a significant reduction in operation time and labor, much less waste produced, and consequently, an overall reduction in operation costs and the potential for higher profitability. In addition, these devices can be relatively easily coupled with other analytical techniques without major investments or manipulation, as is the case of the use of liquid-phase microextraction (DLLME) techniques for the determination of analytes of relevant interest [18]. In this scenario, it is reasonable to consider that solid-phase extraction techniques can also be incorporated into microfluidic devices as has been the case for liquid–liquid extraction systems [19,20]. Miniaturization of such systems maintains their efficiency and at the same time allows avoiding the use of organic solvents for sample pre-treatment, which might limit the durability of the device materials.

Despite these benefits, microfluidics devices have shown some drawbacks. In the case of analytical systems, perhaps the most important is the inherent inefficient mixing resulting from the prevalence of laminar flow. This has been addressed by incorporating complex channel geometries (e.g., serpentines and chambers with circular or semicircular features) that usually improve the contact between the fluids and promote forced diffusive convection. The convex alignment of semicircular elements produces a flow pattern that improves mixing. One channel geometry that falls in such category and that has been recently considered to enable applications in wastewater treatment is the toroidal [21]. In this regard, a torus microreactor exhibits a “coupled mobility matrix” that offers a resistance to flow throughout the translational propulsion of the geometry, around a central rotation point [22,23].

Based on the performance of micromixers and microreactors with curved channels, we hypothesized that a torus-type microreactor could potentially improve the possible interactions between an analyte dispersed in a mobile aqueous phase and the active sites of a solid phase packed within the device. On this basis, we decided to conduct an efficient flow-through static preconcentration process of arsenic within the microreactor by making use of the well-known iron oxide nanoparticles (IONPs) as a solid microphase. IONPs have been already explored as potent adsorbents of metallic analytes with little to no impact on surface charges, pH, ionic concentrations, or temperature [24]. Either bare or functionalized IONPs such as hematite, goethite, ferrihydrite, maghemite, and magnetite (Fe_3_O_4_) have been widely employed for heavy metal extraction purposes [25]. IONPs have been particularly useful in the extraction of different chemical forms of arsenic [26,27,28,29,30,31,32], due to their ability to be easily isolated (collected) upon extraction with the aid of magnetic fields.

The preferred analytical strategy for the detection of an analyte is to make use of a certain device alone; however, this approach fails to provide the required sensitivity, since the collected amount of analyte might be insufficient for in-line detection. This suggests that a pre-concentration stage may be needed to reach the analyte levels [16] required for the subsequent analysis with a separate technique. In the particular case of heavy metals, one of the techniques that can be used is atomic absorption spectroscopy, which not only provides superior sensitivity but also can operate with very small sample volumes (from 2 up to 40 µL). This is advantageous, as the eluted extract (rich in arsenic) from the IONPs is in the same range.

This study was therefore dedicated to implementing a previously introduced toroid microfluidic device packed with magnetite nanoparticles as analyte adsorbents to pre-concentrate heavy-metal-containing samples before their analysis via atomic graphite furnace absorption spectroscopy [33]. The performance of the device compared with that of other geometries was first studied in silico with the aid of COMSOL Multiphysics^®^. As the case study, the device was evaluated with model aqueous samples containing arsenic. All in all, the developed analytical technique could be included in the realm of micro-solid-phase extraction coupled to graphite furnace detection (μ-SPE–GF-AAS) methods.

## 2. Results and Discussion

### 2.1. Multiphysics Simulations of Arsenic Retention on Solid Nanoparticles

Figure 1 shows the particle tracing results, which indicate that the devices were able to retain the MNPs in close proximity to the magnet. Figure 1a,b show that after 10 s, the device was only able to retain a few MNPs, which is also evidenced by the large number of MNPs that came out of the device after the same time (888 particles, or 74% of the total particles).

On the other hand, Figure 1c,d show the corresponding results for the one-loop torus microreactor. Compared with the other device, the number of particles retained by the magnet was significantly higher. Likewise, the number of particles coming out of the device was much lower (97 particles, or 8.04% of the total particles). Based on its superior particle retention performance, we selected the one-loop torus microreactor for further experimental testing.

### 2.2. Characterization of the MNPs

The characterization of the nanoparticles was carried out by powder X-ray diffraction (XRD) using an X-ray diffractometer (Model Empyrean from PANanalitical, Almelo, The Netherlands) (Figure 2). Data collection was performed using Co Kα (λ = 1.7890 nm) radiation (step time, 10 s; step size, 0.065°; 2θ angular range = 5–95°). The size of the particles was also determined by small-angle X-ray scattering (SAXS) (Model Empyrean from PANanalitical, Almelo, The Netherlands) in the same apparatus. Figure 2 shows the diffractogram of the synthesized MNPs in comparison with that reported in the literature and the ICDD PDF database file (International Center for Diffraction Data: PDF file number 72-2303) [34,35,36,37]. The diameter of the MNPs, determined by SAXS, approached 45 nm on average.

### 2.3. Impact of pH on Arsenic Adsorption

To determine the optimal adsorption pH for arsenic on MNPs, 5 mL of a 50 μg·L^−1^ arsenic stock solution was placed in different test tubes followed by the addition of 1.0 mL of 0.01 M sodium acetate/acetic acid buffer solution. Nitric acid and sodium hydroxide were used to adjust the pH to 2, 3, 4, 5, and 6. In the same way, a 0.01 M solution of potassium hydrogen carbonate was employed to adjust the pH between 7 and 9.

Phosphate buffers were avoided, as some fine-structure background signals corresponding to the P–O diatomic molecule might have been present and absorb in the vicinity of the arsenic absorption wavelength (193.696 nm), which was not observed when using carbonate buffers (Figure 3).

Then, 300 µL (4.89 mg) of the synthesized MNPs suspension was added to each tube, followed by vortexing for 5 min. The MNPs enriched with arsenic were separated with the aid of a neodymium magnet, and finally, the arsenic content in each supernatant solution was measured by GF-AAS. This experiment was carried out in triplicate.

Figure 4a shows that arsenic adsorption was favored at pH values close to 6, with the relative retention approaching 98% of the mass contained in a 50 μg·L^−1^ standard. This relative retention percentage was determined by the difference between the initial arsenic concentration and its relative ratio in the remaining supernatant (Equation (1)).
(1)$$\%\;Retention=\left(C_{0}-\frac{C_{suprenatant}}{C_{0}}\right)\times 100\%$$

On the other hand, a significant reduction in the percentage of arsenic retention was observed at pH above 8, which could be attributed to some interfering anions from the buffer solutions used to adjust the alkalinity of the aqueous medium. According to previous studies, the adsorption of arsenic on magnetite should be independent of the pH for values between 4 and 10 [38,39]. Therefore, this result demonstrates that a rigorous study of the chemical components that might interfere with the adsorption process is required before using this preconcentration methodology for the eventual quantification of real samples.

### 2.4. Desorption of Arsenic from the MNPs

Desorption studies were conducted to determine the best arsenic desorption agent and its optimal concentration. Sodium hydroxide and ammonium hydroxide were used as desorption reagents in concentrations ranging from 0.001 M to 5 M. For this purpose, 10 mL of 25 µg L^−1^ arsenic standard solution was adjusted to pH 6 and mixed with 8.0 mg of MNPs. After manual stirring, the mixture was left to rest for one hour followed by three washes with deionized water aided by a magnet. The MNPs were enriched with arsenic and were resuspended again in 250 µL of deionized water. Separately, 300 µL of each NaOH or NH_4_OH solution was tested as a desorption agent by mixing it with 10 µL of the arsenic-enriched MNPs. The mixture was stirred manually for a few seconds and left to stand for 30 min; then the measurement of the arsenic content in the supernatant was carried out.

Figure 5 shows that the desorption of arsenic from the MNPs was enhanced by increasing the pH of the medium, as evidenced by the increased amount of arsenic recovered in the supernatant. The absorbance of each supernatant eluted solution was compared with that produced by a mixture (homogenized by manual agitation) made with 300 µL of deionized water and 10 µL of the enriched MNPs suspension and directly injected into the graphite furnace. This allowed us to take advantage of the graphite furnace capacity for direct analysis of solid samples in suspension (slurry sampling). Importantly, despite the spectral proximity with one of the secondary iron absorption lines, no significant interference was observed with that of arsenic (Figure 3).

The highest recovery percentages were obtained at NaOH concentrations of 3 and 5 M, approaching about 60% of the total recovery. Sodium hydroxide showed higher efficiency compared to ammonium hydroxide, most likely due to a more effective exchange effect on the active sites of magnetite.

### 2.5. µ-SPE–GF-AAS Microextraction Method

For the µ-SPE procedure, the MNPs needed to be magnetically retained in the device’s loop. This was achieved by infusing the device with 10.0 mL of diluted suspension containing 6 mg of magnetite (prepared from the initial stock (16.3 mg·mL^−1^)) at a rate of 14 mL per hour to avoid material losses. After the MNPs were retained in the device, 10.0 mL of a 1.0 μg·L^−1^ solution of arsenic (pH adjusted to 6 with 100 μL of carbonate buffer 0.01M) was infused at the same rate. Subsequently, 5.0 mL of deionized water was infused to remove the free arsenic. Exhaustive retention of arsenic was verified by measuring the element levels in the eluted solution and comparing their concentrations with the limit of detection (LOD) (i.e., 2.216 μg·L^−1^). Further, the method also reduced the typical “characteristic mass” determined by direct measurement from 53.66 pg to 0.88 pg, greatly improving the detection sensitivity. Similar results in terms of sensitivity were found using 10.0 mL of a 2.0 μg·L^−1^ solution of arsenic (obtaining a characteristic mass of 0.83 pg). The whole process is summarized in Table 1.

### 2.6. Evaluation of the Preconcentration Factor for the Measurement of Total Arsenic (TAs) by GF-AAS

The extracting power of the method (MF–µMSPE) was determined at sub-trace levels of the analyte. A preconcentration factor for the method was determined by introducing 20.0 mL of a solution at a very low arsenic concentration (i.e., 1 µg·L^−1^, which was below the LOD) into the device, following steps 1–3 according to the procedure described above. For analyte desorption, 1.8 mL of NaOH 3 M was used, and 10 fractions (200 µL each) eluted consecutively were then collected at the device’s outlet (Figure 6), showing a significant increase in the corresponding arsenic signal.

Compared with the standard arsenic solution (100 ppb), the first alkaline eluate (200 µL) accounted for 61.17% of it, which corresponded to a preconcentration factor of 61. This factor was defined as the ratio between the analyte sensitivities of the two methods when operating at the same initial concentration.

Considering other methods, this factor appears similar (Table 2). However, the proposed coupling strategy exhibited a higher reproducibility (3–6%). Moreover, its LOD was superior to that of HPLC–ICP–MS approaches [40] and comparable with that of HG–FAAS-based methods [38]. Finally, it was possible to carry out the exhaustive cleaning of the nanoparticles using 5 mL of a 2% HCl solution.

## 3. Materials and Methods

### 3.1. Instrumentation

All the arsenic measures were carried out whit an HR–CS–AAS, CONTRAA800-D, atomic absorption spectrophotometer (Analytik Jena, Jena, Germany), equipped with a continuum source consisting of a xenon lamp, a high-resolution monochromator (DEMON), and a 200 pixels diode-array detector. This equipment has the advantage that the optical system itself carries out the BG correction in the vicinity of the analyte signal, in this case at the principal and more sensitive 193.696 nm arsenic working wavelength. The absorbance measurement was determined according to the integrated area, by adding the signals registered in 3 pixels corresponding to the central one (pixel 101) and its immediate adjacent (CP ± 1).

High-purity argon at 2.0 L·min^−1^ was used as the operation inert gas along with pyrolytically coated graphite tubes with an integrated L′vov platform (Analytik Jena Part No. 407-A81.025, Jena, Germany). The tubes were arranged by default in a transverse layout. The employed furnace temperature program is shown in Table 3. We used 12.5 µg Pd as a matrix modifier to improve the arsenic sensitivity, while avoiding losses of the analyte by evaporation during the stages before atomization.

A pH/ORP/Temperature meter, Model PT-380 (BOECO, Hamburg, Germany) equipped with a basic electrode BA 25, Noryl plastic shaft and gel electrolyte was used for pH adjustment. The solutions were pumped into the device by a pulseless infusion single-syringe pump (KDS-100, W.P. Instruments, Holliston, MA, USA).

### 3.2. Reagents and Solutions

All reagents used were analytical-grade. The arsenic standard solution (1.0 g·L^−1^) and concentrated ammonia (30% as NH_3_) were purchased from PanReac AppliChem, Barcelona, Spain. Nitric acid (65% RE, Pure) was purchased from Carlo Erba, Milano, Italy. The palladium graphite matrix modifier (10.0 g·L^−1^) was a commercial solution (Pd(NO_3_)_2·_2H_2_O in 13–20% nitric acid) from Sigma-Aldrich, Darmstadt, Germany. Water was obtained from a Milli-Q deionization system (resistivity of approximately 18 MΩ·cm). All solutions and arsenic standards were prepared fresh before the experiments by direct dilution in deionized water.

### 3.3. Synthesis of the Solid Extractant

The magnetite nanoparticles (MNPs) were synthesized based on the co-precipitation method as described by Mascolo et. al. [34] with slight modifications. To avoid the formation of both maghemite, γ-Fe_2_O_3_, and hematite, α-Fe_2_O_3_, all processes were performed under nitrogen bubbling to displace some oxygen from the synthesis medium. Initially, 100.0 mL of deionized water was heated to 80 °C under a permanent nitrogen flow. Subsequently, 1.00 g of FeCl_2_·4H_2_O and 2.72 g of FeCl_3_·6H_2_O were added, so that the molar ratio of Fe (II) to Fe (III) in the mixture was 1:2. After dissolution, 4.0 mL of concentrated ammonium hydroxide (NH_4_OH) was slowly and progressively added until the pH was raised to 10 to obtain a black precipitate that was stirred for 15 more minutes. The magnetization of the MNPs was verified by collecting them using a permanent neodymium (NdFeB) magnet (30 × 10 × 4 mm block, 1.4 Tesla of field intensity). The MNPs were thoroughly washed with 20 mL of deionized water five times. The supernatant was easily removed by taking advantage of the MNPs magnetism. The obtained MNPs were resuspended again in 40.0 mL of deionized water, obtaining a final suspension (16.3 mg·mL^−1^) with a pH of 7.9.

### 3.4. Modeling of the Microtorus Reactor

The arsenic retention on the MNPs was studied with a particle tracing approach in the COMSOL Multiphysics 6.0^®^ software (COMSOL Inc., Stockholm, Sweden). The fluid was considered under a laminar regime governed by the Navier–Stokes equations according to the conservation of momentum (Equation (2)) and the continuity equation for the conservation of mass (Equation (3)).
ρ(u⋅∇)U = ∇⋅[−pI + μ (∇U + (∇U)^T)] + F(2)
ρ∇⋅U = 0(3)
where µ represents the fluid dynamic viscosity, u the fluid velocity, p the fluid pressure, ρ the fluid density, I the identity matrix, and F the external forces applied on the system. Particle transport was modeled with the aid of the particle tracing module according to the second Newton’s law (Equation (4)).
(4)$$F_{t}=\frac{d\left(m_{p}v\right)}{dt}$$
where *F_t_* corresponds to the sum of all forces acting on the particles, *v* represents the particle velocity, and $m_{p}$ is its mass. The drag force was considered according to (Equation (5)).
(5)$$F_{d}=\frac{1}{\tau_{p}}m_{p}\left(u-v\right),$$
where $\tau_{p}$ is defined by equation (Equation (6))
(6)$$\tau_{p}=\frac{\rho_{p}d_{p}^{2}}{18\;\mu }$$
where $\rho_{p}$ is the particle density, $d_{p}$ is the particle diameter, and μ is the fluid viscosity.

Finally, the magnetophoretic force on the particles exerted by a magnetic field applied was modeled according to (Equation (7)).
(7)$$F_{map}=2\pi r_{p}^{3}\mu_{0}\mu_{r}\frac{\mu_{\mathrm{rp}}-\mu_{r}}{\mu_{\mathrm{rp}}+2\mu_{r}}\nabla H^{2}$$
where $\mu_{r}$ is the relative fluid permeability, $r_{p}$ is the particle radius, $\mu_{\mathrm{rp}}$ is the relative particle permeability, and *H* is the applied magnetic field. The magnetic field was calculated by solving the Maxwell’s equations (Equations (8) and (9)).
(8)$$H=-\nabla V_{m}$$
∇⋅B = 0(9)

In these equations, $V_{m}$ corresponds to magnetic scalar potential, and *B* is the magnetic flux density given by (Equation (10)).
(10)$$B=\mu_{0}\mu_{r}H+B_{r}$$
where $\mu_{0}$ is the vacuum permeability, $\mu_{r}$ is the relative permeability, and $B_{r}$ is the remnant flux density.

The simulations to solve the set of equations of the model were conducted via a bidirectionally coupled particle tracing study with an MUMPS solver. The boundary conditions for these simulations are shown in Figure 7. The computational domain for system 1 (Figure 7a) was meshed with 141,982 domain elements and 9920 boundary elements, while the computational domain for system 2 (Figure 7b) was meshed with 74,941 domain elements and 6618 boundary elements. These meshing levels allowed convergence. The boundary conditions imposed were the drag and the magnetophoretic forces acting on the entire computational domain representing the microfluidic system. In addition, the input of 200 particles to the system every 0.1 s for 0.5 s and the zero magnetic scalar potential at the edge of the computational domain were imposed.

The model assumed a neodymium magnet and water as the channel fluid. All the simulation parameters are summarized in Table 4.

### 3.5. Device Fabrication and Multiphysics Simulations

Figure 7b shows the computational domain of the one-loop torus microreactor as built in COMSOL Multiphysics 6.0^®^ (COMSOL Inc., Stockholm, Sweden). This geometry was chosen to evaluate whether by maximizing the interaction between the MNPs confined to the loop and the flowing arsenic solution, it was possible to improve the solid–liquid extraction efficiency. The device was manufactured by assembling three sheets of polymethylmethacrylate (PMMA) 3 mm thick and with a 75 × 25 mm area. The microchannels (1 mm deep) were engraved using a laser cutting system, Speedy 100, 60 W (TROTEC, Marchtrenk, Austria), followed by gluing them together by applying a thin layer of 96% (*v*/*v*) ethanol and maintaining the assembly under constant pressure for 8 min at 105 °C. Finally, commercially available fittings were inserted in the device inputs and outputs to facilitate further hose connection. Once the devices were manufactured, a constant flow of water was infused through the device by using a syringe pump to check for possible leaks.

A 9 mm-diameter orifice was drilled in all the layers to place a permanent neodymium (NdFeB) magnet (8 mm–diameter cylinder, 0.35 Tesla of field intensity), destined to attract the MNPs and retain them in the device’s loop.

## 4. Conclusions

In this study, it was possible to couple a microfluidic device conditioned with a solid microphase extraction process (MF–µMSPE) to a technique such as atomic absorption to further increase its high sensitivity. In this case, the device was used for the detection of arsenic at sub-trace levels and below the usual limit of detection (LOD) when using the graphite furnace modality (GF-AAS). Compared with some reported SPE pre-concentration methods, automation instead of a batch procedure introduced several important potentialities in terms of simplicity in the treatment (in situ) of the sample and reproducibility of the results (3–6%). Our methodology allowed a relatively good enrichment, since the microreactor geometry assured an efficient contact between the phases in a reduced extraction time. Using a 20 mL volume of standard, or an aqueous sample, a preconcentration factor close to 60 could be achieved using only 200 µL of eluent solution. This factor could be increased even by taking only the first 100 µL. It was determined that 5 M NaOH is an effective and economical eluent for the desorption of the retained arsenic from the magnetite surface of the nanoparticles.

Other important benefits should be highlighted, such as the use of magnetite as a solid extractant, whose synthesis is easy and fast, which showed a high power of concentration of the analyte on its naked, not functionalized, surface. Functionalization, associated with the variation of other factors such as the pH of the aqueous medium, could be carried out to improve the specificity of the extraction process or allow the application to other analytes, potentially reusing the magnetite nanoparticles as the solid extractant. Washing with 5 mL of 2% HCl represents an economic alternative and is time-efficient.

## Figures and Tables

**Figure 1 molecules-27-06198-f001:**
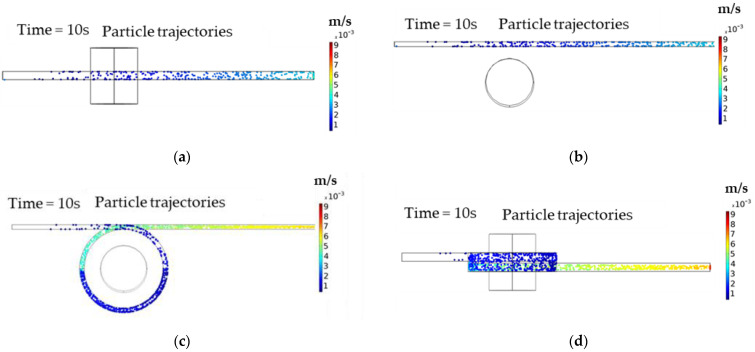
Results of MNPs retention according to the particle tracing simulations. The particles are displayed in color according to their velocities. (**a**) Side view of the particles trajectory after 10 s in the microfluidic device without a loop, (**b**) top view of the particles trajectory after 10 s in the microfluidic device without a loop, (**c**) side view of the particles trajectory after 10 s in the one-loop torus microreactor, and (**d**) top view of the particles’ trajectories after 10 s in the one-loop torus microreactor.

**Figure 2 molecules-27-06198-f002:**
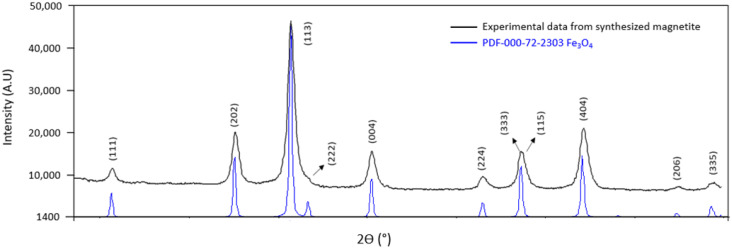
XRD patterns of the synthesized magnetite nanoparticles compared with those in a database (ICDD data: PDF file number 72-2303).

**Figure 3 molecules-27-06198-f003:**
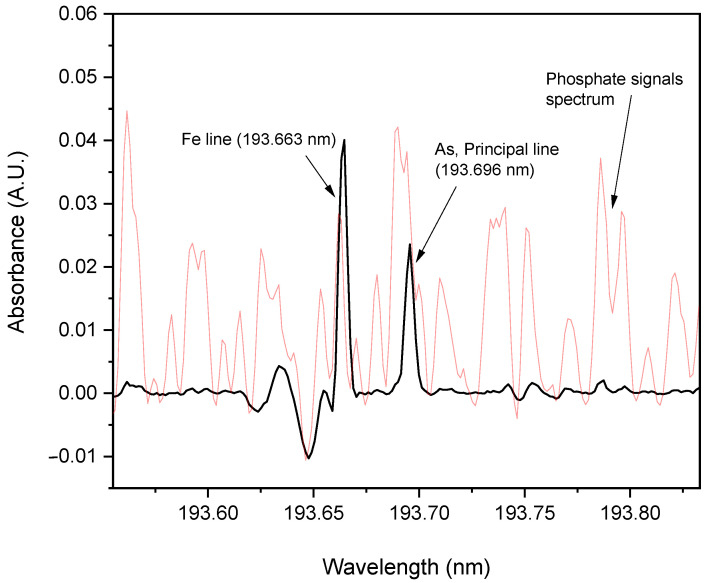
Average arsenic atomic absorption spectrum in the vicinity of 193.696 nm. Direct injection of 5 µL of MNPs slurry (1 mg·mL^−1^), enriched with arsenic, in the presence of 0.01 M of carbonate buffer, evidencing the absence of any possible interfering molecular phosphate signals (spectrum corresponding to potassium dihydrogen phosphate, 0.1 M, overlapped in red).

**Figure 4 molecules-27-06198-f004:**
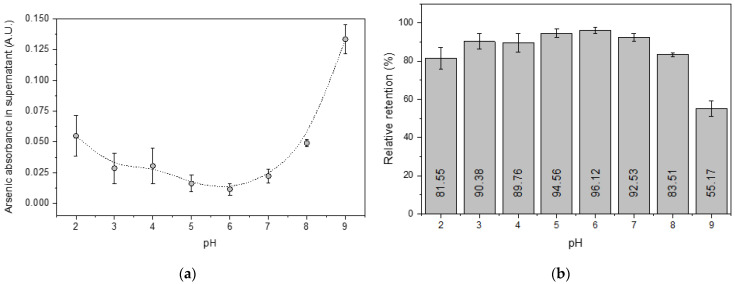
Influence of pH on arsenic retention by the MNPs from 5 mL of a 50 μg·L^−1^ arsenic stock solution. (**a**) Concentration of arsenic in the collected supernatants and (**b**) respective retention percentages. Mean values (three measurements); the error bars represent the corresponding standard deviations.

**Figure 5 molecules-27-06198-f005:**
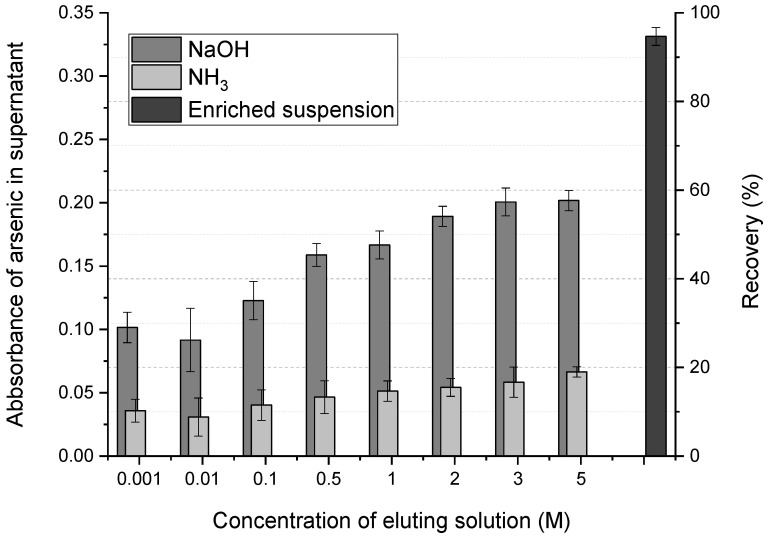
Evaluation of NaOH and NH_3_ for arsenic desorption from the MNPs. Arsenic absorbance in the supernatant, after carrying out the desorption with 300 µL of the alkali and 10 µL of an MNPs suspension enriched in arsenic. Mean values (three measurements); error bars represent the corresponding standard deviations.

**Figure 6 molecules-27-06198-f006:**
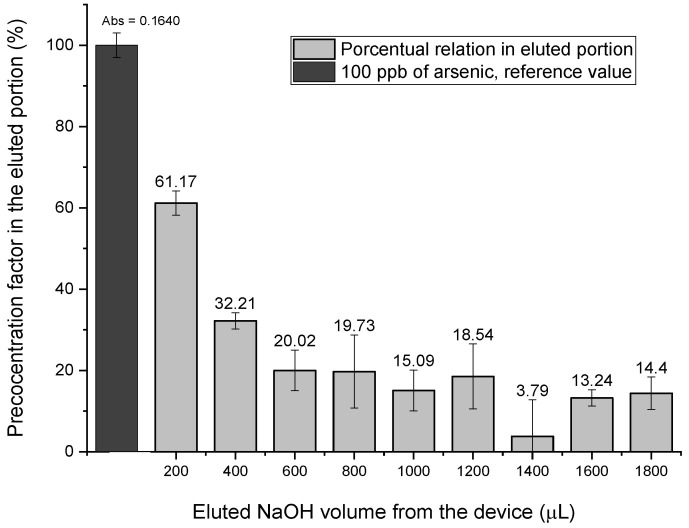
Percentage of eluded arsenic from the device. Mean values (three measurements); error bars represent the corresponding standard deviations.

**Figure 7 molecules-27-06198-f007:**
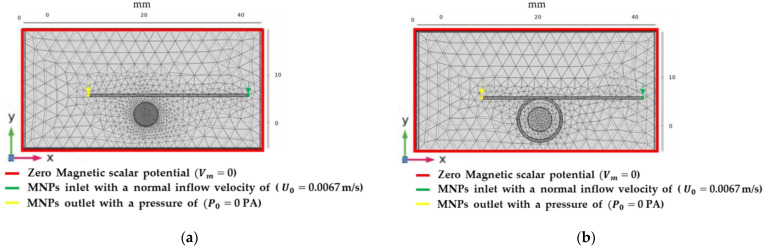
Boundary conditions for the simulations. (**a**) Torus microreactor with one loop for arsenic retention on solid nanoparticles with one loop, (**b**) microfluidic system for arsenic retention on the MNPs. For both systems, the drag force and the magnetophoretic force were imposed on the microfluidic channel, where the inlet is shown in green, and the outlet in yellow, and a zero magnetic scalar potential was assumed at the edge of the computational domain.

**Table 1 molecules-27-06198-t001:** General operation procedure for the µ-SPE–GF-AAS system.

Step	Reagents	Volume (mL)	Flow Velocity (mL·h^−1^)
Charging loop	Magnetite suspension (0.6 mg·mL^−1^)	10	15
Standard load	Arsenic standard, with known concentrations	10/20	30
Washing	Deionized water	3	30
Desorption	Sodium hydroxide, 3 mol·L^−1^)	0.200	15
MNPs removal and cleaning	Hydrochloric acid 2% (*v*/*v*)	5	30
Regeneration of the medium	Deionized water	3	30
Measurement of the arsenic signal in 20 µL of the alkaline extract by GF-AAS.			

**Table 2 molecules-27-06198-t002:** Comparison of some analytical methods for the preconcentration of arsenic.

Sample	Arsenic Species	Enrichment Method	Instrumentation Method	LOD (ng L^−1^)	RSD%	Preconc. Factor	Ref.
Garlic	Inorganic As	Ionic liquid-assisted multiwalled carbon nanotube-dispersive micro-solid phase extraction (IL-MSPE)	ETAAS	7.1	4.8–5.4	70	[40]
Seawater	Total As	SPE	HG-FAAS	0.02–0.03	5.3	-	[41]
Rice	Inorganic As	GPE	LC–HG–in situ DBDT–AFS	0.05	<2	11	[42]
Fish	Inorganic As	Ionic imprinted polymer-solid-phase extraction (IIP-SPE)	HPLC–ICP–MS	0.32–0.39	12	50	[43]
Rice	Inorganic As	In situ quaternary ammonium salt solid-phase extraction (ISQAS- SPE)	FI–HG AAS	0.04	5.5	17	[44]
Water	As(III), As(V), MMA	Micro-solid-phase extraction (µ-SPE)	ETAAS	0.02	5.4	98	[30]
Water, vegetables and rice	As(III)	UA–μSPE on a magnetic ion-imprinted polymer	HG-AAS	0.003	3.21	120	[45]
Water	Inorganic As	MF–µMSPE	GF-AAS	0.033	3–6	60	This work

**Table 3 molecules-27-06198-t003:** Graphite furnace temperature program. High-purity argon was used in all steps except during the atomization (and reading) when the gas flow was turned off.

Step	Temperature (°C)	Heating Ramp (°C·s^−1^)	Hold Time (s)	Argon Flow Rate (L·min^−1^)
Drying 1	80	6	10	2.0
Drying 2	110	3	5	2.0
Pyrolysis 1	300	80	5	2.0
Pyrolysis 2 ^a^	1100	350	5	2.0
Gas adaptation ^b^	1100	0	5	Stopped
Atomization	2400	2400	4	Stopped
Cleaning	2500	500	4	2.0

^a^ The modifier was injected alone, running steps 1 and 2, and cooling the atomizer, before the standard injection. ^b^ The argon flow was adjusted to the atomization conditions.

**Table 4 molecules-27-06198-t004:** Parameters for the simulation of arsenic retention on the MNPs.

Parameter	Value	Units
$\mathrm{Water}\;\mathrm{relative}\;\mathrm{permeability}\;(\rho_{w}$)	1	Dimensionless
$\mathrm{Air}\;\mathrm{relative}\;\mathrm{permeability}\;(\rho_{A}$)	1	Dimensionless
$\mathrm{Neodymium}\;\mathrm{relative}\;\mathrm{permeability}\;(\mu_{N}$)	1.05	Dimensionless
$\mathrm{Velocity}\;\mathrm{inlet}\;(u_{0})$	0.0067	m/s
$\mathrm{Pressure}\;\mathrm{outlet}\;(P_{O}$)	0	Pa
$\mathrm{Particle}\;\mathrm{density}\;(\rho_{p}$)	5180	kg/m³
$\mathrm{Particle}\;\mathrm{radius}\;(r_{p}$)	45	nm
$\mathrm{Particle}\;\mathrm{relative}\;\mathrm{permeability}\;(\mu_{p}$)	5000	Dimensionless
Remnant flux density (Br)	0.35	T

## Data Availability

Not applicable.

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
