# Peer review of "Magnetic Torus Microreactor as a Novel Device for Sample Treatment via Solid-Phase Microextraction Coupled to Graphite Furnace Atomic Absorption Spectroscopy: A Route for Arsenic Pre-Concentration"

_molecules, 2022, doi:10.3390/molecules27196198_

Round 1

Reviewer 1 Report

Manuscript Number: molecules-1908856

1. I stated that a new microreactor based on torus geometry is investigating the investigation of model aqueous samples containing Arsenic. At the same time, the performance of the manuscript device compared to other geometries was first examined in silico with the help of COMSOL Multiphysics.

2. This manuscript is to develop a new microreactor based on torus geometry and demonstrate a remarkable improvement in the analyte adsorption process. The manuscript is interesting in this respect and was used for the first time for As determination. The ace is made by performing retention on the surface of a magnetic sorbent. At the same time, a simulation theoretical study using this geometric design shows a remarkable improvement in the analyte adsorption process.

Title:  
Magnetic torus microreactor as a novel device for sample treat- 2 ment via solid-phase microextraction coupled to graphite fur- 3 nace atomic absorption spectroscopy: A route for Arsenic pre- 4 concentration.  

3. The Manuscript is well written, clear and easy to read.

4. I recommend that it is compared it with the results of the study to this manuscript has just been published in Molecules Journal, “Determination of Ultra-Trace Cobalt in Water Samples Using Dispersive Liquid-Liquid Microextraction Followed by Graphite Furnace Atomic Absorption Spectrometry(Molecules 2022, 27, 2694. https://doi.org/10.3390/molecules27092694)

5. This manuscript includes the use of a new microreactor based on torus geometry and the examination of model aqueous samples containing Arsenic.

6. There are only µg∙L-1 NH4OH,  mg∙mL-1  typos in the text.

This article is acceptable.

Author Response

Dear reviewer, thank you very much for your suggestions, we show below the respective responses and comments to your valuable recommendations:

Manuscript Number: molecules-1908856

  1. I stated that a new microreactor based on torus geometry is investigating the investigation of model aqueous samples containing Arsenic. At the same time, the performance of the manuscript device compared to other geometries was first examined in silico with the help of COMSOL Multiphysics.

Response: We appreciate the reviewer’s positive comments.

  1. This manuscript is to develop a new microreactor based on torus geometry and demonstrate a remarkable improvement in the analyte adsorption process. The manuscript is interesting in this respect and was used for the first time for As determination. The ace is made by performing retention on the surface of a magnetic sorbent. At the same time, a simulation theoretical study using this geometric design shows a remarkable improvement in the analyte adsorption process.

Response: We appreciate the reviewer’s positive comments.

Title:  Magnetic torus microreactor as a novel device for sample treat- 2 ment via solid-phase microextraction coupled to graphite fur- 3 nace atomic absorption spectroscopy: A route for Arsenic pre- 4 concentration.  

  1. The Manuscript is well written, clear and easy to read.

Response: We appreciate the reviewer’s positive comments.

  1. I recommend that it is compared it with the results of the study to this manuscript has just been published in Molecules Journal, “Determination of Ultra-Trace Cobalt in Water Samples Using Dispersive Liquid-Liquid Microextraction Followed by Graphite Furnace Atomic Absorption Spectrometry(Molecules 2022, 27, 2694. https://doi.org/10.3390/molecules27092694)

Response: We appreciate very much the reviewer´s suggestion; however, we believe that the suggested work corresponds to a liquid phase microextraction methodology, which differs significantly from the type of microextraction presented in our work (solid phase microextraction). However, in the introduction, we cite your recommended article as an application of miniaturization methodologies for sample treatment, since it is a relevant and pertinent work in that regard (line 59).

  1. This manuscript includes the use of a new microreactor based on torus geometry and the examination of model aqueous samples containing Arsenic.

Response: We certainly agree with the reviewer.

  1. There are only µg∙L-1 NH4OH,  mg∙mL-1  typos in the text.

Response: All the µg∙L-1 and mg∙mL-1 typos in the manuscript were fixed

Dear reviewer, Thank you very much for your valuable suggestions

Sincerely, The authors.

Reviewer 2 Report

Dear Authors,

Thank you for such interesting research. I suggust, the article requires major revision. Here is few major points:

1. Please, provide the description of matrix effects and non-selectivr absorbance.

2. The research should soundes better if you provide results of the QC sample analysis (QC high, medium, low).

3. Figure 6 is not clear and requires another presentation form. Maybe, you should provide the preconcntration factor at one of the axis? Right now it looks strange for me, because it sounds like "Recovery" but this is preconcentration. 

4. Figure 3. "SecondaRy line"? Please, check the caption.

5. Did the Authors have any opinion why a significant reduction in the percentage of Arsenic retention can be seen at pH above 8?

6. Tolerance at all figures should be checked. Have you validated that values on your figures? There is no information about repetitions for each point in the manuscrupt.

Warm regards

Author Response

Dear reviewer, thank you very much for your suggestions, we show below the respective responses and comments to your valuable recommendations:

Dear Authors,

Thank you for such interesting research. I suggust, the article requires major revision. Here is few major points:

  1. Please, provide the description of matrix effects and non-selectivr absorbance.

Response: We wish to present the work as a sample preconcentration strategy for eventually designing a stage prior to the determination of arsenic by graphite furnace. For this reason, our work did not focus on a formal quantification of real wastewater samples, but rather on evaluating water solutions with known Arsenic concentrations. Therefore, recovery or standard addition studies were not carried out, and consequently, we disregarded non-selective absorbance for our studies. Perhaps, table number 2 could have been misguiding as we used the word “determination” in the title (line 249). We changed it to enhance the clarity of the table.

  1. The research should soundes better if you provide results of the QC sample analysis (QC high, medium, low).

Response: We agree with the reviewer’s suggestion and in fact, we also tested a higher concentration of 2 ppb, obtaining a similar concentration factor. However, in principle, we decided to use 1 ppb as it is a lower and more notable concentration to reflect the pre-concentration power of the device. Based on your important recommendation; however, we rewrote part of section 2.5. to explicitly incorporate this suggestion (line 220).

  1. Figure 6 is not clear and requires another presentation form. Maybe, you should provide the preconcntration factor at one of the axis? Right now it looks strange for me, because it sounds like "Recovery" but this is preconcentration.

Response: We agree with the reviewer’s suggestion and therefore improved Figure 6 by changing the axes’ description according to the suggested recommendations.

  1. Figure 3. "SecondaRy line"? Please, check the caption.

Response: We eliminated the word secondary in Figure 3 for clarity. The term “secondary” is a formalism that has been extensively used by the atomic absorption community to avoid referring to the main line (highest sensitivity). This iron line appears in the vicinity of the arsenic signal and does not interfere with the measurement of the arsenic signal because of the high resolution of the detector (HR-CS-AAS).

  1. Did the Authors have any opinion why a significant reduction in the percentage of Arsenic retention can be seen at pH above 8?

Response: With respect to the significant reduction in As retention above pH 8, we included an improved discussion in lines 162 to 164. In this new paragraph, we explained that the reduction could be likely attributed to the presence of some interfering anions from the buffer solutions used to adjust the alkalinity of the aqueous medium. This considering that according to previous studies, the adsorption of arsenic on magnetite should be independent of pH for values 4 and 10.

  1. Tolerance at all figures should be checked. Have you validated that values on your figures? There is no information about repetitions for each point in the manuscrupt.

Response: As pointed out by the reviewer, we failed to include the standard deviation of the data (three replicas) and only included the average values. We included a description of how we calculated the deviation and clarify it in each figure.

We, the authors, hope to have given satisfactory corrections and clarification to all your suggestions and recommendations.

Sincerely,

The Authors

Round 2

Reviewer 2 Report

I suggest the manuscript could be accepted in present form.